# Neurofilaments as Emerging Biomarkers of Neuroaxonal Damage to Differentiate Behavioral Frontotemporal Dementia from Primary Psychiatric Disorders: A Systematic Review

**DOI:** 10.3390/diagnostics11050754

**Published:** 2021-04-22

**Authors:** Vincent Davy, Julien Dumurgier, Aurore Fayosse, Claire Paquet, Emmanuel Cognat

**Affiliations:** 1Cognitive Neurology Center, AP-HP, GH Saint-Louis Lariboisière Fernand-Widal, F-75010 Paris, France; vcnt.davy@gmail.com (V.D.); julien.dumurgier@aphp.fr (J.D.); claire.paquet@inserm.fr (C.P.); 2Sorbonne Université, F-75010 Paris, France; 3Université de Paris, UMRS 1153, INSERM, F-75010 Paris, France; aurore.fayosse@inserm.fr; 4Université de Paris, UMRS 1144, INSERM, F-75010 Paris, France

**Keywords:** neurofilaments, blood biomarkers, behavioral frontotemporal dementia, neuropsychiatry, diagnosis, single molecular array

## Abstract

The behavioral variant of frontotemporal dementia (bvFTD) is a clinical syndrome resulting from various causes of neuronal demises associated with frontotemporal lobar degeneration. Symptoms include behavioral and personality changes, social cognitive impairment, and executive function deficits. There is a significant clinical overlap between this syndrome and various primary psychiatric disorders (PPD). Structural and functional neuroimaging are considered helpful to support the diagnosis of bvFTD, but their sensitivity and specificity remain imperfect. There is growing evidence concerning the potential of neurofilaments as biomarkers reflecting axonal and neuronal lesions. Ultrasensitive analytic platforms have recently enabled neurofilament light chains’ (NfL) detection not only from cerebrospinal fluid but also from peripheral blood samples in FTD patients. In this short review, we present recent advances and perspectives for the use of NfL assessments as biomarkers of neuroaxonal damage to differentiate bvFTD from primary psychiatric disorders.

## 1. Introduction

Frontotemporal dementia (FTD) is a heterogeneous group of neurodegenerative disorders characterized by a combination of behavioral changes, social cognitive impairment, language and memory impairments, and executive function deficits. These clinical symptoms result from the prominent degeneration of neurons in the frontal and temporal lobes associated with diverse underlying pathology [1]. FTD is divided into three major clinical syndromes: The behavioral variant (bvFTD) [2] and the two language variants referred to as semantic or non-fluent primary progressive aphasias [3]. The diagnosis of bvFTD is challenging because cognitive impairment may be absent or subtle in the early stages, and initial symptoms, including behavioral disinhibition, apathy, lack of empathy, dietary changes, and compulsions, may be suggestive of primary psychiatric disorders (PPD). This clinical overlap with late-onset PPD leads to a high rate of misdiagnosis at the initial stages of bvFTD. Indeed, in a large retrospective study, about 50% of bvFTD patients received a prior diagnosis of a psychiatric disorder [4]. This led international experts to recently publish consensus recommendations to distinguish bvFTD from PPD [5].

Thus, biomarkers with the potential to accurately discriminate between late-onset PPD and bvFTD are needed to help early diagnosis and to initiate appropriate patient management. CSF and blood neurofilament proteins, which are axonal structural proteins, have emerged as biomarkers of axonal damage in various neurological disorders [6]. Moreover, neurofilament light chains’ (NfL) detection from peripheral blood samples is now possible with ultrasensitive analytic platforms [7]. This convenient and less invasive option than CSF assessments could be especially suited for neuropsychiatric differential diagnostic.

In this short review, we first focus on the clinical overlap between bvFTD and relevant PPD. We then present the biomarkers used in current practice for the diagnosis of bvFTD and their respective limitations, especially regarding FTD/PPD differential diagnosis. Finally, after a short introduction to the broader use of neurofilaments in neuropsychiatry, we review and discuss recent studies exploring NfL as biomarkers for the differential diagnosis between bvFTD and late-onset PPD.

## 2. Methods

We performed a systematic search in accordance with the Preferred Reporting Items for Systematic Reviews and Meta-Analyses guidelines. We identified all published articles from database inception to April 2020 on the Medline and Cochrane databases using the following Medical Subject Heading (MeSH) terms: “frontotemporal dementia” OR “frontotemporal lobar degeneration” OR “primary psychiatric disorder” OR “bipolar disorder” OR “depression” OR “psychosis” AND “neurofilament” AND “blood” OR “serum” OR “plasma” OR “cerebrospinal fluid” OR “biomarker”. Papers were first screened based on titles and abstracts (*n* = 145) and then preselected articles (*n* = 40) were read in full. Articles were included in the systematic review using the following criteria: The study (1) was available in English; (2) reported primary data; (3) included individuals diagnosed with FTD or PPD; (4) had included at least 70% of bvFTD patients in the FTD group or primary data were available for bvFTD; (5) quantified NfL in human CSF, plasma, or serum using the prominent NfL antibody (mAb47:3, UmanDiagnostics, Umeå, Sweden) either in commercial or homemade kits; and (6) reported NfL concentration as mean and standard deviation, or the median and interquartile range (IQR) or confidence interval. This search and inclusion criteria are described in the flowchart (Figure 1) and resulted in 29 articles selected for the systematic review.

Medians and IQR were converted to means and standard deviation following the methodology presented by Hozo et al. [8]. Data extracted from the included studies are summarized in Appendix A. Because the retrieved data were not abundant enough to perform a comparative analysis, we made a pooled forest plot analysis. We also examined papers cited in the selected articles and included additional references based on their relevance regarding the scope of this paper. 

## 3. The Clinical Overlap between bvFTD and PPD

Major depressive disorder (MDD) is one of the most frequent psychiatric disorders. It is characterized by sadness associated with a wide range of clinical symptoms, including lack of interest, social withdrawal, and decreased motivation. Moreover, severe major depression often causes impaired cognitive processing [9]. In some patients, these cognitive and behavioral symptoms are prominent over sadness, a clinical situation that can result in increased difficulty to perform a valid differential diagnosis with bvFTD. This is especially true in late-life depression, which is characterized by less subjective sadness, less insight, and more cognitive symptoms [10].

Bipolar affective disorders were the second most common psychiatric misdiagnosis in bvFTD patients in a retrospective study [4]. Indeed, bvFTD and mania share common symptoms such as socially inappropriate behaviors, excessive joviality, impulsivity, hypersexuality, and overspending. These symptoms, which are constitutive of early behavioral disinhibition, are included in clinical criteria of both bipolar disorders and bvFTD.

Negative symptoms of schizophrenia include marked apathy, paucity of speech, and emotional flattening, which are also observed in bvFTD. While symptoms usually start at a younger age in schizophrenia than in bvFTD, late-onset schizophrenia has been described. In addition, the cognitive deficits described at the initial stage of bvFTD, such as deficits in attention, working memory, executive function, and processing speed, are frequently observed in patients with schizophrenia [11]. The two diseases also share similar social cognitive deficits and limited insight. Finally, florid psychosis was reported as an atypical initial presentation of bvFTD. Especially, late-onset psychosis at the initial stage of the disease seems particularly frequent in patients carrying the C9ORF72 mutation. Results from two different cohorts of FTD patients with C9ORF72 mutations revealed a similar rate of about one-third of psychosis at initial presentation (12/32 in a British cohort [12] and 10/33 in an Italian cohort [13]). In these studies, persecutory delusions, paranoid ideations, and both visual and auditory hallucinations were the most frequent symptoms.

Obsessive–compulsive disorder (OCD) is less frequently reported as a misdiagnosis, but compulsions have been described as the sole initial manifestation of bvFTD [14]. Stereotyped and ritualistic behaviors are one of the clinical bvFTD diagnosis criteria [2]. They range from simple stereotypies to more elaborate rituals mimicking the compulsions observed in OCD. Of note, marked anxiety is frequently lacking in bvFTD patients with only OCD. However, the DSM V definition of OCD highlights the fact that OCD can be present without the insight that OCD beliefs are untrue [15]; OCD with poor insight can be hard to differentiate from bvFTD compulsive behaviors.

It is worth noting that the assessment of family history can also be misleading. Between 35% and 50% of FTD patients have a family history of FTD or amyotrophic lateral sclerosis (ALS) [16]. Even if the inheritance mechanisms are unclear, the familial aggregation in bipolar affective disorders [17] and schizophrenia [18] is also high. In addition, FTD patients harboring the C9ORF72 mutation have a higher frequency of psychiatric disorders in their family history [19]. Altogether, a study analyzing the risk factors of prior PPD diagnosis in patients with early neurodegenerative diseases showed that positive family history of psychiatric disorders biased physicians toward psychiatric rather than neurodegenerative etiologies [4].

Finally, a recently described FTD-like entity may be responsible for peculiar diagnostic challenges. Cohort studies have identified patients fulfilling diagnostic criteria for possible bvFTD and showing neither neuroimaging supportive features nor clear cognitive deterioration over time. In this subgroup, called “bvFTD phenocopy”, patients are predominantly men, are less impaired in executive functions or social cognition, and have normal lifespan [20]. Some of these individuals have very slowly progressive forms of bvFTD secondary to C9ORF72 [21]. A recent longitudinal study reported that a majority do not fulfill criteria of probable bvFTD after up to 21 years of follow-up [22]. This raises the question of undiagnosed mild PPD, such as atypical residual mood disorders or low-grade psychotic disorders, although much uncertainty remains as to the nature of this condition.

## 4. Biomarkers to Support the Diagnosis

Apart from genetically proven cases, definitive biomarkers for the diagnosis of bvFTD are lacking. Routine CSF biomarkers (amyloid-β, t-tau, and p-tau) are helpful to distinguish FTD from AD and are listed in the exclusionary criteria of bvFTD [2]. Currently, as no specific CSF biomarker pattern has been identified in FTD [23], they are of no help to distinguish FTD from PPD.

The 2011 consensus criteria [2] introduced imaging biomarkers (structural or functional imaging) as possible tools to increase diagnostic probability. Disproportional atrophy in the medial frontal, orbital–insular, and anterior temporal regions on brain MRI are observed in bvFTD patients [22]. However, a neuropathologically confirmed cohort showed that 50% of bvFTD patients were lacking these typical MRI features, reflecting the low sensitivity of structural brain imaging in bvFTD patients, especially at the early stages of the disease [24]. This is in line with the idea that clinical and functional abnormalities may precede structural imaging changes in the disease course, as shown by brain functional imaging. However, a study assessing the added value of [18 F]-FDG-PET in patients with suspected bvFTD and normal structural MRI reported a sensitivity of only 47%, while the specificity was high (92%) [25]. A report from the Late-Onset Frontal Lobe Syndrome Study (an observational prospective follow-up of patients with late-onset frontal behavioral change consisting of apathy, disinhibition, or compulsive/stereotypical behavior) found that the specificity of [18 F]-FDG-PET might be lower in bvFTD patients with initial psychiatric presentation [26].

Thus, there is a global unmet need for new biomarkers to facilitate an earlier and more accurate diagnosis of bvFTD, and this is even more crucial for PPD/bvFTD differential diagnosis issues.

## 5. Neurofilaments as Emerging Biomarkers in Neurological Disorders

Neurofilaments (Nf) are neuronal-specific cytoskeleton filaments. They are composed of heteropolymers of three subunits, namely, neurofilament heavy, medium, and light (NfH, NfM, and NfL) that assemble to form long and thin intermediate filaments. Nf are expressed throughout the whole cell and are especially enriched in the axonal region. Thus, axonal injury causes the release of Nf proteins into the extracellular fluid, the amount depending on the extent of damage [27]. Interestingly, Nf subunits levels can be measured in CSF and, more recently, in peripheral blood using high-sensitivity detection techniques such as the single molecule array (Simoa^TM^) assays [28].

Consequently, numerous studies have demonstrated that levels of Nf proteins, especially NfL in CSF and plasma, are increased in a wide range of neurological diseases [6]. Both CSF and serum NfL, for instance, have been extensively studied in multiple sclerosis in which they revealed to be interesting biomarkers for assessment of disease activity, prognosis, and drug response [29,30]. CSF NfL have also been studied in various neurodegenerative disorders. Two cohort studies demonstrated that CSF NfL levels can effectively differentiate Parkinson’s disease from atypical parkinsonian disorders [31,32]. In ALS, CSF NfL levels are especially high, which aids differential diagnosis and correlates with disease extent and prognosis [33,34,35]. Among neurocognitive degenerative disorders, a mild elevation of NfL has been reported in Alzheimer’s disease [36], but levels are consistently higher in FTD [37]. In addition, both CSF and serum NfL levels seem to correlate with the severity of the disease, either measured by survival rates [38] or frontal lobe atrophy rate [39] in FTD patients.

## 6. Neurofilaments Used to Differentiate PPD from bvFTD

Contrary to neurodegenerative diseases such as bvFTD, PPD are not considered to be associated with prominent neuroaxonal damage. However, one study found a slightly elevated CSF NfL level in bipolar patients [40], whereas CSF NfL levels in different primary psychiatric disorders (including bipolar disorder, major depressive disorder, and schizophrenia) did not differ from controls in another study [41]. Similarly, in one study exploring serum NfL in patients with major depressive disorder receiving electroconvulsive therapy, baseline serum NfL concentrations did not differ between patients and healthy controls [42].

Very few studies directly compared NfL concentrations in bvFTD and PPD to estimate the comparative degrees of NfL changes and the usefulness for differential diagnosis. In two studies comparing several CSF biomarkers of neurodegeneration in bvFTD and PPD, CSF NfL appeared to provide the best discriminative performance either alone [43] or in combination with total Tau [44]. Because NfL detection in plasma is now available, recent studies also examined the usefulness of NfL in this setting. Al Shweiki et al. [45] compared serum NfL levels in 20 bvFTD patients and 50 psychiatric patients (11 schizophrenia, 11 bipolar, and 28 depression patients) and found 85% sensitivity and 78% specificity with a cutoff of 23.7 pg/mL to differentiate bvFTD from PPD. No significant differences were observed between serum NfL levels in psychiatric patients and controls. A second study by Katisko et al. [46] with 66 bvFTD patients and 34 PPD showed that serum NfL could differentiate bvFTD and PPD with 79% sensitivity and 85% specificity using a cutoff level of 19.9 pg/mL. It is worth noting that the cutoffs reported in these two studies were in the same range as the one previously described in studies comparing bvFTD patients and healthy controls [47]. Indeed, as shown in Figure 2 and Figure 3, the studies published so far show relatively low variability of NfL levels in both CSF and serum in the groups of interest, with values in the same range in PPD patients and in controls that are much below those observed in FTD patients and with little apparent overlap.

## 7. Unsolved Questions

These results were highly encouraging, but some points remain to be clarified in future studies. First, several studies, such as the one from Al Shweiki and colleagues, focused on younger patients (both PPD and bvFTD). As serum NfL levels steadily increased with aging (6), it would be important to confirm the results in older patients. Second, most studies included mainly or only bvFTD patients classified as probable according to the current diagnostic criteria, that is, with imaging indicative of bvFTD (86% of the 286 patients with detailed information available). As mentioned above, new biomarkers would be especially needed in possible bvFTD cases when supportive imaging biomarkers are lacking. In addition, as Nf levels have been suggested to directly correlate with the intensity of neurodegeneration, the added diagnostic value of Nf to structural and functional imaging markers in FTD at the early stages has yet to be evaluated. This would be made possible by analyzing NfL levels at baseline in patients with possible bvFTD/late frontal syndrome enrolled in prospective longitudinal studies with sufficient follow-up or pathologic confirmation. Finally, the use of serum NfL as a screening tool in primary care settings would imply the establishment of standardized measurement procedures and consensus cutoff values.

## 8. Conclusions

Due to a significant symptomatic overlap, differentiating bvFTD from PPD is a frequent diagnostic challenge. Biomarkers used currently in clinical practice to facilitate this diagnostic process have limited sensitivity. Misdiagnosis of bvFTD leads to ineffective and potentially harmful treatments, delays in organizing proper support, and increased family stress [48]. Disease-modifying treatments for FTD are currently in the research pipeline and some are already under investigation in clinical trials. Thus, there is an urgent need for earlier and more accurate diagnosis of bvFTD, especially when the initial presentation is equivocal. In late-onset neuropsychiatric presentation, the level of CSF and/or serum neurofilaments could be a convenient measure to assess whether or not a neurodegenerative process is already in progress.

## Figures and Tables

**Figure 1 diagnostics-11-00754-f001:**
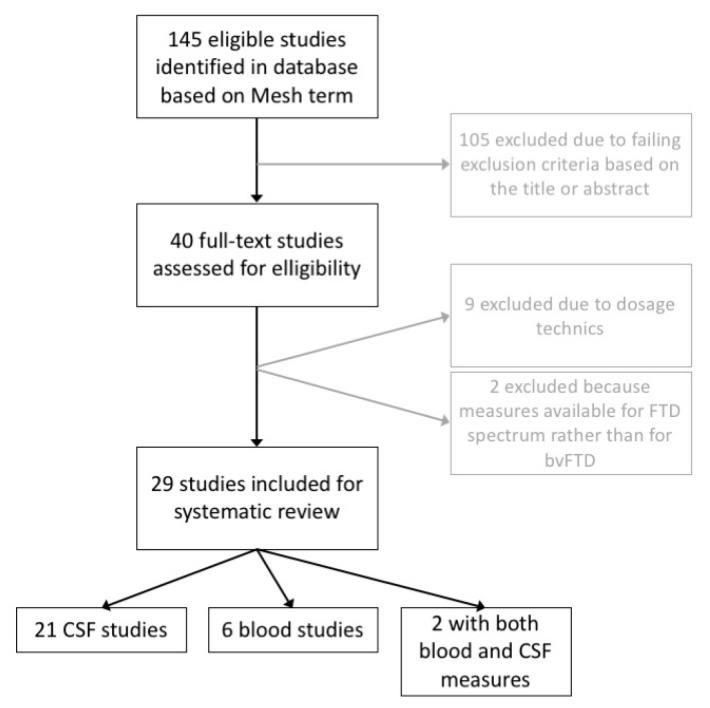
Flowchart of the selection process.

**Figure 2 diagnostics-11-00754-f002:**
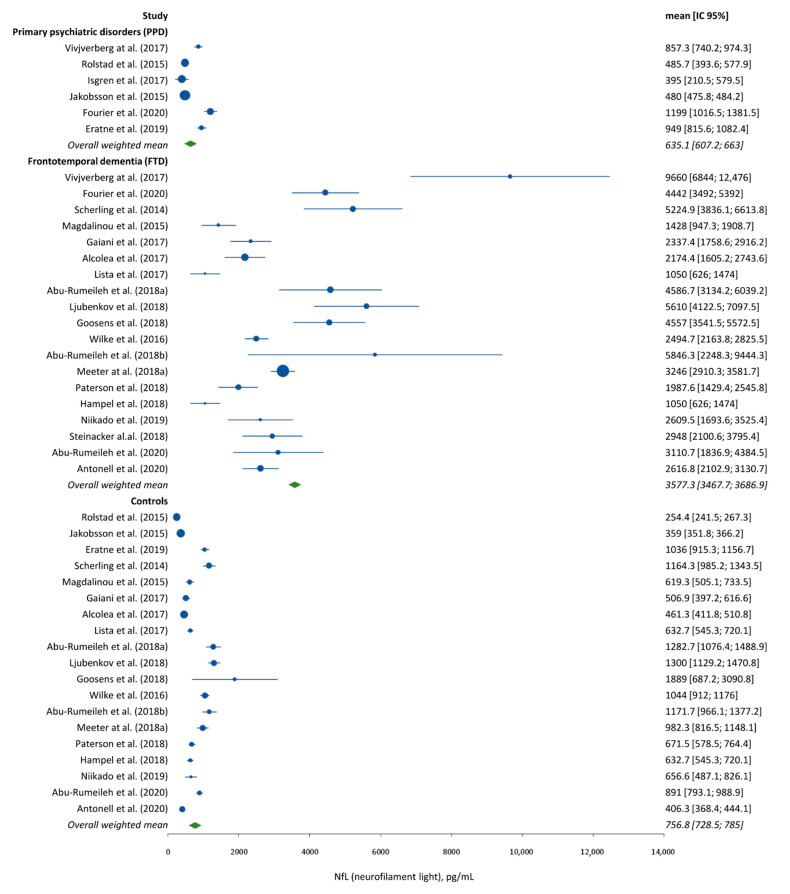
Forest plot analysis of CSF NfL levels in FTD, PPD, and control patients.

**Figure 3 diagnostics-11-00754-f003:**
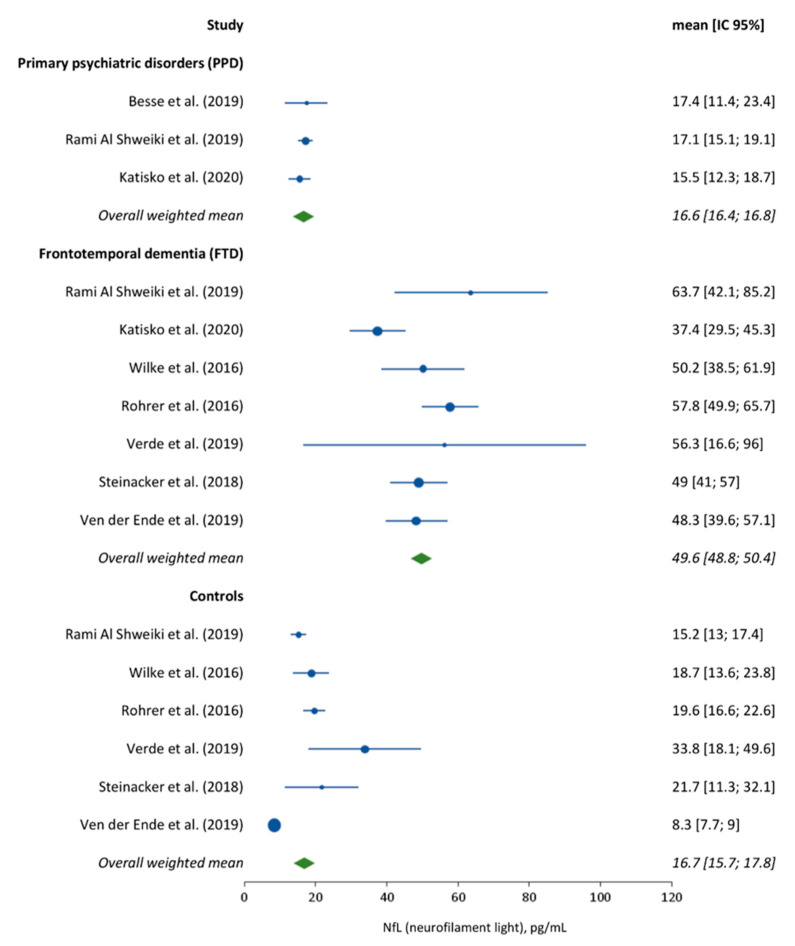
Forest plot analysis of serum NfL levels in FTD, PPD, and control patients.

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
