# Peer review of "Neurofilaments as Emerging Biomarkers of Neuroaxonal Damage to Differentiate Behavioral Frontotemporal Dementia from Primary Psychiatric Disorders: A Systematic Review"

_diagnostics, 2021, doi:10.3390/diagnostics11050754_

Round 1
Reviewer 1 Report
the review is clear, well written and usefull
areas of strength: to support differential diagnosis at early stages, specifically helping to distinguish behavioral symptoms related to neurodegeneration from the ones associated with psychiatric disorders.Author Response
We are very grateful to our reviewer for his highly encouraging comments.
For our reviewer information, we upload the revision version modified in accordance with the comments form our three reviewers.
Reviewer 2 Report
This is a very interesting short review on the implementation of neurofillaments (light chains-Nf-L) as emerging biomarkers in the differential diagnosis of the behavioural variant of frontotemporal dementia (bvFTD) and late onset primary psychiatric disorders (PPD).
This is a very hot topic in the intersection point between neurology and psychiatry, where significant overlap in clinical symptoms between the two groups of disorders exist, while biomarkers are lacking.
The paper is emphasizing on Nf-L as a biomarker useful in providing evidence in favor of bvFTD in the clinical setting bvFTD vs. PPD, through forest plot analyses of CSF and serum Nf-L levels in FTD, PPD and control subjects.
The review of Davy et al is methodologically efficient, well organized structurally, written in a comprehensive way and I suggest to be published.
Minor:
C9ORF72, check to appear homogeneously all over the text.
Author Response
We are very grateful to our reviewer for his complementary comments.
We homogeneized C9ORF72 writing throughout the manuscript.
For our reviewer information, we upload the revision version modified in accordance with the comments form our three reviewers.
Reviewer 3 Report
Dear editor
Thank you for allowing me to review the present manuscript entitled “Neurofilaments: emerging biomarkers of neuroaxonal damage to differentiate behavioral frontotemporal dementia from psychiatric disorders”.
The manuscript is well-written and the research question is adequate and interesting. Authors clearly have profound insight in the clinical disease presentations and their diagnostic challenges. I, on the other hand, have insufficient insight in these aspects and cannot evaluate this part of the manuscript sufficiently.
From a methodological and structural point of view the manuscript has some major flaws. In the methods section it states to have performed a search according to the PRISMA guidelines. However, the manuscript clearly does not follow the PRISMA check list. There is no clear description of the search criteria, the study selection process is in sufficiently described, there is no bias assessment or quality assessment of the included studies. There is no aggregated summary of the findings. As such, the result of the literature review is not clear to me.
Altogether, the manuscript provides an interesting review on clinical aspects and challenges of frontotemporal dementia and psychiatric disorders but contains little in terms reviewing the literature on neurofilaments as biomarkers for these conditions. Especially, it suffers from severe problems in reporting on the literature search and its results. I would suggest a major revision were authors either adhere to the guidelines for systematic reviews or re-writes the manuscript as a narrative review.
Author Response
We thank our reviewer for his comments that helped us improve our manuscript on very important methodological aspects.
From a methodological and structural point of view the manuscript has some major flaws. In the methods section it states to have performed a search according to the PRISMA guidelines. However, the manuscript clearly does not follow the PRISMA check list. There is no clear description of the search criteria, the study selection process is in sufficiently described, there is no bias assessment or quality assessment of the included studies. There is no aggregated summary of the findings. As such, the result of the literature review is not clear to me.
We acknowledge this criticism. While we actually followed the prisma process to perform our litterature review, we did not provide sufficient information, especially in the methods section of the manuscript to substantiate this statement.
Altogether, the manuscript provides an interesting review on clinical aspects and challenges of frontotemporal dementia and psychiatric disorders but contains little in terms reviewing the literature on neurofilaments as biomarkers for these conditions. Especially, it suffers from severe problems in reporting on the literature search and its results. I would suggest a major revision were authors either adhere to the guidelines for systematic reviews or re-writes the manuscript as a narrative review.
In order to correct this issue, we rewrote the methods section of the paper:
- we detailed the exact search terms according to the PRISMA guidelines
- we detailed more precisely the search criteria
- we described the study selection process with all the inclusion criteria
- we illustrated the study selection process with a flowchart
In addition, we provide the reader with an aggregated summary of the findings through a supplementary table.
We hope that the revised manuscript will now seem suitable for publication to our reviewer.
Round 2
Reviewer 3 Report
Dear editor and authors
I would like to thank you for allowing me to review this interesting paper.
The authors have adequately addressed my concerns and I recommend publication in Diagnostics.
As a last suggestion, I would recommend a slight revision of the title to include the term “systematic review and meta-analysis” as also recommended by the PRISMA guidelines.
Author Response
We are glad that our answers were found appropriate by our reviewer.
To take into account his/her last proposal, we have updated the title of the paper to :
Neurofilaments as emerging biomarkers of neuroaxonal damage to differentiate behavioral frontotemporal dementia from psychiatric disorders: a systematic review.
We think that adding "meta-analysis" would be kind of an overstatement. However, we let the final decision to also add this term to the title or not to our editor.